# Synthesis of Well-Defined Gold Nanoparticles Using Pluronic: The Role of Radicals and Surfactants in Nanoparticles Formation

**DOI:** 10.3390/polym11101553

**Published:** 2019-09-24

**Authors:** Marina Sokolsky-Papkov, Alexander Kabanov

**Affiliations:** 1Center for Nanotechnology in Drug Delivery Eshelman School of Pharmacy, UNC-Chapel Hill, Chapel Hill, NC 27514, USA; msokolsk@email.unc.edu; 2Laboratory of Chemical Design of Bionanomaterials, Faculty of Chemistry, M.V. Lomonosov Moscow State University, Moscow 119992, Russia

**Keywords:** gold nanoparticles, Pluronic block copolymers, polymeric micelles

## Abstract

Synthesis of gold nanoparticles (GNP) by reacting chloroauric acid (HAuCl_4_) and Pluronic F127 was thoroughly investigated. The rate of reduction of HAuCl_4_ and the yield and morphology of GNP strongly depended on the concentration of the reactants and sodium chloride, as well as pH and temperature. Upon completion of the reaction heterogeneous mixtures of small GNP of defined shape and Pluronic aggregates were formed. GNP were separated from the excess of Pluronic by centrifugal filtration. Under optimized conditions the GNP were small (ca. 80 nm), uniform (PDI ~0.09), strongly negatively charged (ζ-potential −30 mV) and nearly spherical. They were stable in distilled water and phosphate-buffered saline. Purified GNP contained ~13% by weight of an organic component, yet presence of polypropylene oxide was not detected suggesting that Pluronic was not adsorbed on their surface. Analysis of the soluble products suggested that the copolymer undergoes partial degradation accompanied by cleavage of the C–O bonds and appearance of new primary hydroxyl groups. The reaction involves formation of free radicals and hydroperoxides depends on the oxygen concentration. GNP did not form at 4 °C when the micellization of Pluronic was abolished reinforcing the role of the copolymer self-assembly. In conclusion, this work provides insight into the mechanism of HAuCl_4_ reduction and GNP formation in the presence of Pluronic block copolymers. It is useful for improving the methods of manufacturing uniform and pure GNP that are needed as nanoscale building blocks in nanomedicine applications.

## 1. Introduction

Gold nanoparticles (GNP) have attracted considerable attention due to their potential applications in biology and medicine as drug delivery and sensory devices. Control of size, shape, and colloidal stability of the GNP is very important for these applications. A main synthetic route for GNP involves reduction of chloroauric acid (HAuCl_4_) using sodium citrate or sodium borohydrate [1]. Although tuning the feed ratio of gold and sodium salts can control the size of GNP to some extent, the resulting GNP usually are not stable in aqueous solution. Their colloidal stabilization can be further achieved by surface functionalization with various SH-containing species and represents an additional synthetic task. As alternative, stable GNP can be synthesized in the presence of certain water-soluble polymers (polyethyleneimine, poly(*N*-isopropyl-acrylamide-*N*,*N*-dimethylaminoethyl-acrylamide [2,3,4,5], or polyethers, such as poly(ethylene oxide) (PEO) or Pluronic, a triblock copolymer of PEO and poly(propylene oxide) (PPO) (PEO-PPO-PEO) that can adsorb on the GNP surface and form a protective stabilizing layer in situ [6,7,8,9,10]. Furthermore, the polymer layer around the GNP can be utilized for additional biomedical applications such as incorporation of drugs and utilized as theranostic drug delivery systems. Interestingly, these polymers concurrently act as reducing agents. 

As true for any synthetic process of practical significance, identifying the role of all reaction components is extremely important upon synthesis of GNP. For amine containing polymers the reducing ability of the polymer can be clearly attributed to the amines. However, in the case of polyethers the question is more complicated and despite extensive investigations remains uncertain. Numerous studies characterized interaction of Pluronic block copolymers with metal colloids as well as the role of the block copolymer in HAuCl_4_ reduction and colloidal particles formation. The correlations between the Pluronic composition and the characteristics of the formed Au nanoparticles were evaluated to tune the size and the shape of the nanoparticles as well as to determine the role of unimers and/or micellar assemblies in the process of the reduction reaction and nanoparticles formation [7,8,10,11,12]. However, analysis of the previously reported data suggested no straightforward correlation between various Pluronic characteristics, reduction kinetics and GNP formation. For example, increase in polymer concentration was reported to result in the formation of either more heterogeneous [11] or more uniform GNP [9]. Sakai and Alexandridis postulated a three-step reduction and GNP formation process attributing the driving force for nanoparticles formation to block copolymers-Au^0^ interaction and the nanoparticles growth to reduction of Au^3+^ on the surface of the nanoparticle’s seeds. Polte et all [13], reported on the GNP formation in the presence of Pluronic F127 showing burst reduction of Au^3+^ followed by GNP nucleation and growth due to Au^0^ coalescence with the nuclei. Kohlbrecher et al. [14] showed that GNP formation in the presence Pluronic P85 was dictated by HAuCl_4_/polymer ratio, however most block copolymer remained unassociated with GNPs. 

The objective of this work was to: 1) establish a robust and repeatable synthetic procedure route for uniform and well defined GNP; 2) advance understanding of the role of the block copolymers in the reduction reaction; 3) examine interactions between Pluronic and GNP; and 4) delineate environmental factors (pH, NaCl concentration, temperature) governing the particle formation. The ultraviolet–visible (UV–vis) spectroscopy and dynamic light scattering (DLS) were used to examine the rates of HAuCl_4_ reduction and particle formation. Formation of free radicals and hydroperoxides during the reaction was analyzed by electron paramagnetic resonance (EPR) spectroscopy. The GNP morphology was analyzed by transmission electron microscopy (TEM) and atomic force microscopy (AFM). The chemical composition and structure of the products of the reaction were analyzed by Fourier transform infrared spectroscopy (FTIR), nuclear magnetic resonance (NMR), gel permeation chromatography (GPC), and thermo-gravimetric analysis (TGA). Under optimized conditions we produced small, uniform, negatively charged, and nearly spherical GNP that were fully separated from the block copolymer and were stable in aqueous media. The study provides a valuable insight into the mechanism of HAuCl_4_ reduction and GNP formation and could be useful for improving methods of manufacturing of uniform and pure nanoparticles.

## 2. Materials and Methods

### 2.1. Materials

Pluronic F127 ((PEO)_200_(PPO)_65_(PEO)_200_, Mw 12600), Pluronic P104 ((PEO)_54_(PPO)_63_(PEO)_54_, Mw 5900), Pluronic P85 ((PEO)_52_(PPO)_40_(PEO)_52_, Mw 4600) and Pluronic F88 ((PEO)_208_(PPO)_39_(PEO)_208_, Mw 11400), were supplied by BASF corporation. Hydrogen tetrachloroaurate (III) trihydrate (HAuCl_4_·3H_2_O), PEO (Mw 8000), sodium chloride (NaCl), hydrochloric acid (HCl), sodium hydroxide (NaOH), 4-acetamidophenol (AAP), horseradish peroxidase (HRP), and diethylenetriaminepentaacetic acid (DTPA) were purchased from Sigma-Aldrich, St. Louis, MO. 1-Hydroxy-3-methoxycarbonyl-2,2,5,5-tetramethylpyrrolidine (CMH), EPR buffer, diethyldithiocarbamic acid sodium salt (DETC), and deferoxamine (DF) were purchased from Noxygen Science Transfer and Diagnostics, Elzach, Germany All reagents were used as received without farther purification. All aqueous solutions were prepared using bi-distilled water (Millipore, Billerica, MA, USA).

### 2.2. Synthesis of GNP

Particles were synthesized as described by Sakai8 with slight modification. Briefly, 1 mL of aqueous stock solution of HAuCl_4_ was added to 1 mL of Pluronic stock solutions, vortexed for 10 s and incubated at desired temperatures until the reaction was completed (the PR band of the optical spectra attained plateau level). In the typical experiment, 1 mL of 12.6% *w*/*w* Pluronic F127 aqueous solution was mixed with 1 mL of 0.5 mM HAuCl_4_ aqueous solution and the reaction mixture was incubated for 2 h at 45 °C. Pluronic F127 solutions with different pH values were prepared by adding 1M NaOH or 1M HCl to stock 12.6% *w*/*w* aqueous solution of Pluronic F127. pH of the solutions before and after mixing with of HAuCl_4_ aqueous solution was measured using Oakton three-point calibration pH meter. Pluronic F127 solutions with various NaCl concentrations were prepared by mixing different volumes of 20% *w*/*w* aqueous solution of Pluronic F127 and 5M NaCl aqueous solutions. Pluronic F127 concentration in the final stock solution was 12.6% *w*/*w*. 

### 2.3. GNP Purification

Particles were purified from excess of the copolymer by centrifugal filtration at 1500 rpm for 30 min at 4 °C (repeated three times). The purified nanoparticles were restored to original volume with distilled water. Particles concentration remained the same before and after purification (no GNP were detected in the filtrate). 

### 2.4. UV–Vis Spectroscopy

HAuCl_4_ reduction and GNP formation in reaction mixtures as well as purified GNP solutions were analyzed by UV–Vis spectroscopy. The measurements were carried out using PerkinElmer Lambda 25 UV–Vis multi-chamber spectrometer. The instrument was operated at spectrum mode with the wavelength interval of 2 nm, and the samples were held in quartz cuvettes of path length 1 cm. Pluronic F127 aqueous solutions of the same concentration were used as reference. 

### 2.5. Physico-Chemical Analysis

#### 2.5.1. DLS Analysis

Effective hydrodynamic diameter (D_eff_) and ζ-potential of F127 solutions, reaction mixtures and purified GNP were determined by DLS using a Zetasizer Nano ZS (Malvern Instruments Ltd., Malvern, UK). All measurements were performed in automatic mode at 25 °C. Software provided by the manufacturer was used to calculate the size, polydispersity indices (PDI), and ζ-potential. All measurements were performed at least in triplicate to calculate mean values ± SD. The reaction mixtures were analyzed as prepared without further dilutions unless specified otherwise.

#### 2.5.2. TEM Analysis

Twenty microliters of a concentrated aqueous solution of the purified GNP (synthesized under optimal conditions as previously described) were deposited on the top of carbon-coated grid for 1 min, the excess of the solution was removed, and the grid was dried for 2 min at RT. In some samples, negative staining (2% of methylamine tungstate solution) was applied (one drop was deposited on the dried grid for 1 min, removed and the grid was dried at RT for 2 min). TEM imaging was performed using a Tecnai GZ Spirit TW device (FEI, Hillsboro, OR, USA) equipped with an AMT digital imaging system (Danvers, MA, USA) operating at 80 kV.

#### 2.5.3. AFM Analysis

Five microliters of an aqueous solution of the purified GNP were deposited onto positively charged aminopropylytriethoxy silane (APS) mica surface for 2 min, washed with water and dried under argon atmosphere. AFM was carried out with MFP-3D microscope (Asylum Research, Santa Barbara, CA, USA) mounted on inverted optical microscope (Olympus, Center Valley, PA, USA) operated in tapping mode. The imaging in air was performed with regular etched silicon probes (TESP) with a spring constant of 42 N/m.

#### 2.5.4. Surface Analysis of GNP and Reaction Products 

To analyze the reaction products, the particles were synthesized by mixing 1 mL of Pluronic F127 aqueous solution (0.63% *w*/*w*) and 1 mL of aqueous HAuCl_4_ solution (0.5 mM)-1:1 molar ratio. Reaction mixture was incubated at 45 °C for 24 h. The nanoparticles were separated from the reaction mixture as described above and the separated filtrate was lyophilized to dryness. To analyze the GNP the particles were synthesized, separated from the reaction mixture and lyophilized to dryness.

#### 2.5.5. FT-IR

The dried reaction products or GNP were analyzed in solid state using a Nicolet 380 FT-IR spectrometer (Thermo Scientific, Waltham, MA, USA). FTIR spectra were recorded in the wavelength range of 4000–400 cm^−1^.

#### 2.5.6. NMR

The dried reaction products or GNP were dissolved in D2O. ^1^H-NMR and ^13^C-NMR spectra were acquired at 25 °C using a Bruker AVANCE III spectrometer (400 MHz, Billerica, MA, USA). Chemical shifts are expressed in parts per million downfield from Me_4_Si as internal standard.

#### 2.5.7. GPC

The number-average molecular mass (Mn), weight-averaged molecular mass (Mw), and molecular mass distribution (Mw/Mn) of Pluronic F127 before and after nanoparticles synthesis were determined by GPC, using Viscotek GPC system equipped with G3000PWXL column and refractive index detector. All measurements were performed at 25 °C with a flow rate of 1 ml/min using distilled water containing 0.02% NaN_3_ as an eluent. The calibration of column was performed using PEO standards (Mw 2–22 kDa, Viscotek, TX, USA).

#### 2.5.8. TGA

TGA was carried out on the purified dried GNP (synthesized under optimal conditions as described above) using a TA Instruments TGA Q500 instrument. After equilibrating the samples at 25 °C, the temperature was ramped at 10 °C min^−1^ to a maximum temperature of 650 °C under a nitrogen purge. Char yields (the mass remaining at the end of the experiment) were recorded at the maximum temperature.

#### 2.5.9. Free Radical and Hydroperoxide Assays

The free radicals and hydroperoxides in the samples were assayed by EPR spectroscopy. These studies used a ‘KDD buffer’ containing EPR Krebse HEPES buffer (99 mM NaCl, 4.69 mM KCl, 2.5 mM CaCl_2_, 1.2 mM MgSO_4_, 25 mM NaHCO_3_, 1.03 mM KH_2_PO_4_, 5.6 mM D-glucose, 20 mM 4-(2-hydroxyethyl)-1-piperazineethanesulfonic acid (HEPES)) and metal chelators, 4.34 mM DETC and 1.85 mM DF. The following samples were studied: 1) 6.3% solutions of F127 (before and after lyophilization); 2) 0.25 mM HAuCl_4_; 3) reaction mixture containing 1) and 2); 4) unpurified GNP dispersion (2 h, 45 °C). To determine free radicals 68.52 μL of sample solutions in bi-distilled water were mixed with 26.6 μL of KDD buffer and 4.88 μL of 4.1 mM CMH spin probe in KDD buffer. To determine hydroperoxides 68.52 μL of sample solutions were mixed with 13.74 μL of 7.28 mM AAP, 7.87 μL of 2.54 mM DTPA, 4.99 μL of 20 U/ml HRP and 4.88 μL of 4.1 mM CMH spin probe (all in KDD buffer) [15]. The final concentrations of the reagents were as follows: 5 mM DETC, 25 mM DF, 1 mM AAP, 0.2 mM DTPA, 1U/mL HRP, and 0.2 mM CMH. Mixtures were incubated for 5 min at room temperature and then 50 μL of these mixtures was loaded into a glass capillary tube and analyzed in Bruker Biospin e-scan EPR spectrometer spectrometer (Bruker). The CMH radical signal was recorded at 37 °C and EPR spectrum amplitude was quantified in arbitrary EPR units. The net hydroperoxide signal was calculated as a difference between the EPR spectrum amplitudes in AAP/DTPA/HRP containing and free KDD buffers.

#### 2.5.10. In Vitro Stability of Purified GNP

Stability of purified GNP was evaluated in distilled water or phosphate-buffered saline (PBS). Particles were synthesized by mixing 1 mL of F127 aqueous stock solution (12.6% *w*/*w*) and 1 mL HAuCl_4_ aqueous stock solution (0.5 mM) followed by incubation of the mixture for 2 h at 45 °C. Reaction mixture was purified from excess of the polymer as described above and the concentrated solution was restored into 2 mL with distilled water or PBS respectively. The solutions were incubated at 37 °C and effective hydrodynamic diameter was measured by DLS as described above. 

## 3. Results

### 3.1. Preliminary Considerations and Choice of a Lead Copolymer

GNP were synthesized in the aqueous media through a reaction of HAuCl_4_·3H_2_O with Pluronic block copolymers. Several block copolymers (Appendix A) were evaluated in order to select one, which can: 1) effectively reduce the Au^3+^ and 2) facilitate the formation of the uniform, stable GNP. These copolymers had different PEO and PPO content, HLB and CMC values. Reactions involving the reduction of Au^3+^ to Au^0^ and formation of the GNP were monitored by the changes in the UV absorption spectra (Appendix A). The greatest differences between the copolymer samples were observed in the position and shape of the PR peak (ca. 530–580 nm.). The particles formed in the presence of F127 displayed relatively narrow PR peak centered between 530 and 550 nm. Moreover, only these particles were stable, while for the rest of the copolymers, the particles precipitated in several hours. Hence, we selected F127 as the lead copolymer in this study. 

### 3.2. Effect of the Temperature on Au^3+^ Reduction and GNP Formation

To evaluate the effects of the temperature, F127 was mixed with HAuCl_4_·3H_2_O and incubated at 25 °C or 45 °C. Immediately after mixing the absorbance intensity at 220 nm decreased sharply (from 3 to 1.9) suggesting that the initial phase of Au^3+^ reduction to Au^0^ proceeded very rapidly at both temperatures. This rapid phase was followed by subsequent slower Au^3+^ reduction as evidenced by a decrease of intensity at 220 nm (Figure 1). The slower phase was temperature dependent and clearly faster at 45 °C than at 25 °C. Altogether, at 45 °C the initial AuCl_4_^−^ peak center shifted to 240 nm corresponding to Au^0^ after 30 min of the incubation, compared to 180 min for 25 °C. This was followed by the phase corresponding to the formation of GNP exhibited by a decrease in the absorbance intensity at 240 nm and increase of absorbance intensity in the PR region. This process was also faster at 45 °C compared to 25 °C. However, despite the differences in the rates of reduction and the particles formation, the final absorbance intensity values both at 240 nm and PR region were the nearly same (Figure 1). Following completion of the reaction the aqueous dispersions of the obtained GNP were characterized by DLS. This method confirmed the formation of the nanoparticles, which were slightly negatively charged. Interestingly, when the reaction mixture was diluted by at least two-fold the particle size (here and below particle sizes are reported as effective diameters, D_eff_), PDI and ζ-potential values sharply decreased (Figure 2). Therefore, we purified the GNP from the excess of F127. The purified GNP represent isolated and relatively uniform structures (Figure 3A,B). Moreover, TEM analysis of the particles did not show any difference between the particles synthesized at 25 °C and 45 °C in either size or shape (Appendix A). In both cases the particles were either close to spherical or have hexagon-like projections. Based on these studies, subsequent synthesis of GNP was carried out at 45 °C, unless otherwise specified. Importantly, under the reaction conditions reported above we were able to produce nearly spherical particles of ca 80 nm (by DLS), which were relatively uniform with a PDI as low as 0.09. The hydrodynamic size and PDI values of purified particles revealed some changes in double distilled water (DDW) and PBS (Appendix A). Thus, the particle size decreased over time from about 70 nm to about 60 nm in both DDW and PBS. The PDI values increased in PBS (to nearly 0.2), with less change observed in DDW. No precipitation was observed during the experiment. The purified particles also did not precipitate in 10% NaCl (1.72 M) although they did exhibit some changes in effective diameters (83.49 ± 3.94 nm vs. 67.18 ± 0.32 nm) and polydispersity (PDI 0.278 ± 0.012 vs. 0.12 ± 0.008).

### 3.3. Effect of HAuCl_4_ Concentration on the GNP Formation and Characteristics

The above-described reaction conditions resulted in formation of relatively uniform spherical nanoparticles. We further examined the effect of HAuCl_4_·3H_2_O concentration on the size, ζ-potential and shape of the unpurified aggregates and purified nanoparticles. The concentration of HAuCl_4_·3H_2_O was varied in the range of 0.05 to 0.25 mM while keeping F127 concentration constant (6.3% *w*/*w*). The PR peak absorbance increased linearly suggesting that more nanoparticles were formed as HAuCl_4_ concentration increased (Figure 4A,B). This was accompanied by the increase in the size and decrease in the PDI of the unpurified aggregates until 0.15 mM HAuCl_4_·3H_2_O concentration was reached. Above this concentration, the size and PDI remained constant (approx. 200 nm and PDI 0.22). Interestingly between 0.1 and 0.15 mM HAuCl_4_·3H_2_O the position of the PR peak shifted from 530 nm to 540 nm indicating the change of the particle shape. Analysis of the purified particles by TEM revealed that their shape was indeed affected (Table 1 and Appendix A). At low HAuCl_4_·3H_2_O concentrations (0.05–0.1 mM) the particles were mostly spherical, small, and relatively uniform by size (TEM). As the concentration increased (0.15–0.2 mM) consistent with the change of the PR peak the triangles and rods appeared among the spherical particles. At the highest concentration (0.25 mM) the rods disappeared and mostly spherical particles with some triangles were formed again. However, in contrast to the effective diameters of the aggregates the effective diameters of the purified GNP practically did not depend on the HAuCl_4_·3H_2_O concentration (Table 1). As already noted, their ζ-potential was strongly negative compared to the ζ-potential of the aggregates, which was low in the absolute value. The ζ-potential of the purified particles or aggregates did not depend on the HAuCl_4_·3H_2_O concentrations.

### 3.4. Effect of F127 Concentration on the GNP Formation and Characteristics

Next, we varied the concentration of F127 while keeping the HAuCl_4_·3H_2_O concentration constant (0.25 mM). The absorbance at the PR peak increased and the position of the PR peak shifted to 530 to 550 nm. as the F127 concentration increased (Figure 4C,D). The effective diameter of unpurified aggregates also increased from about 100 nm to about 300 (Table 2). However, once the GNP were purified their effective diameters did not depend on the copolymer concentration and ranged from 60 to 80 nm while PDI values gradually decreased from about 0.3 to 0.09. Notably, the ζ-potential of the purified particles was strongly negative (−30 to −45 mV) and much lower than of any unpurified particles (−11 to −2.5 mV). TEM analysis revealed some alterations in the shape and heterogeneity of the particles (Table 2, and Appendix A). At lower concentrations of the copolymer (1–3.15 wt %) the purified particles were heterogeneous by both size and shape, representing a mixture of smaller and larger spheres, triangles and rods. As the concentration of F127 increased, the rods disappeared, followed by disappearance of triangles; at 7.5% and 10% F127 relatively homogeneous by size and shape and nearly spherical or cubic particles were formed. 

### 3.5. Relationship between Polymer Aggregation and GNP Formation

Notably, the concentrations of the copolymer in this study were above the critical micelle concentration (CMC) at the given temperatures of the experiment [16]. Under these conditions the copolymer exists in solution in two forms: 1) the micelle aggregates and 2) the single chains (unimers) that are in equilibrium with the micelles [16]. To elucidate the role of the self-assembly of F127 in GNP formation we carried out same reactions but at 4 °C when the micellization of F127 is abolished at every concentration used [16]. At 1–2% F127 there was little reduction of Au^3+^ during the observation time of 2 h (Figure 5). However, the reduced Au^0^ formed some GNP as exhibited by a well-defined albeit small PR peaks. At higher F127 concentrations the reduction of Au^3+^ was nearly complete judging by the shift of the absorption peak to 240 nm. However, the PR peaks were much smaller and less defined compared to the PR peaks observed at the same F127 concentrations at higher temperatures (25 °C and 45 °C), when the copolymer formed micelles. Another evidence for the role of the micelles was seen using non-micelle-forming 10% PEO 8000 as reducing agent at 45 °C. Albeit Au^3+^ reduction appeared to be slower compared to the F127 systems, the GNP still formed. However, these GNP were highly heterogeneous representing assembly of sizes and shapes in contrast to the nearly spherical particles observed at similar concentrations of F127 (Appendix A). This suggests that the F127 micelles play some role in the GNP formation, perhaps serving as the templates for nucleation and/or growth of Au^0^ atoms favoring formation of the spherical particles. Interestingly, we obtained evidence suggesting that the size of the particles and the self-assembly of the copolymer change in the course of the reaction. First, the DLS suggested that addition of HAuCl_4_·3H_2_O (0.25 mM) to F127 solution (5 mM, 6.3%) resulted in the immediate increase of the particle size from 27 nm corresponding to F127 micelles to 102 nm. Subsequently, the particle size continued to increase in the course of the reaction to 160 nm (Appendix A). Moreover, when the HAuCl_4_·3H_2_O solution was carefully added as a layer on top of the previously formed concentrated F127 gel (30 wt%, room temperature), the reaction proceeded at the interface with the GNP forming in the mixed layer, which spread frontally through the entire gel volume. Notably in course of this reaction the gel structure of F127 was disrupted ultimately producing the homogeneous liquid dispersion containing GNP. The color of the formed GNP solution was purple suggesting formation of large, irregular particles. This observation is very interesting since it shows that the HAuCl_4_·3H_2_O is reduced with copolymer present in the upper layer of the gel and the reaction disrupts the interaction between the copolymer chains and destroys the gel matrix.

### 3.6. Effect of pH on the GNP Formation and Characteristics

The size and morphology of GNP depended on the pH of the F127 initial solution. Table 3 presents the results of the GNP synthesis for different pH values before and after mixing the F127 and HAuCl_4_·3H_2_O solutions. 

When the reaction was carried out at acidic conditions (initial pH 2.2 to 4.5) the formed GNP particles were heterogeneous by size and shape as seen by TEM analysis of the purified particles. Moreover, at the lowest pH 2.2 the reduction reaction was considerably decelerated and required 24 h for completion based on the UV spectra analysis (data not shown). On the contrary, as the pH of the initial F127 solution increased and approximated neutral (pH 6.7 to 7.5) and then became alkali (pH 9.0 to 11.5) the particles became more homogeneous by size and shape as determined by TEM. Furthermore, the particle size considerably decreased as suggested by both DLS and TEM analyses (Table 3, Appendix A). It is well known that the TEM particle size is usually less than the effective diameters measured by DLS, which is commonly explained by the hydration and swelling of the particle polymer coatings during the DLS studies [17]. However, the differences between the TEM and DLS diameters in the present case were too strong especially for the purified GNP produced using alkali solutions of the copolymer—e.g., ca. 12 nm by TEM vs. 40–50 nm by DLS. Most likely these differences were due to aggregation of the GNP in aqueous solution during the DLS study. Notably, as determined by DLS the PDI indexes of the purified GNP synthesized using alkali solutions of the copolymer were relatively high while the TEM study suggested that the particles were rather homogeneous by size and shape. Such discrepancy in polydispersity assessments by DLS and TEM are consistent with the particle aggregation in solution. At the same time this aggregation appeared to be restricted as the particle effective diameters measured by DLS remained relatively small. Moreover, for the GNP synthesized using alkali solutions of the copolymer, no significant difference in the size of the purified and non-purified particles was observed and these particles remained stable in dispersion showing no aggregation or precipitation for several weeks. In contrast, the particles formed using acidic solutions of the copolymer exhibited significant differences in size before and after purification and had a tendency to precipitate during first 24 h after their synthesis. Interestingly, the ζ-potentials of both purified and unpurified particles were nearly the same regardless the initial pH used in the synthesis (Table 3).

### 3.7. Effect of NaCl on the GNP Formation

Addition of NaCl to the reaction medium profoundly affected the formation of GNP. However, the results depended on the temperature. Thus, the analysis of the UV spectra of the reaction mixture incubated at 25 °C revealed that Au^3+^ was reduced in the entire range of NaCl concentrations (0.005 M to 1.25 M). However, below 0.05 M NaCl this was accompanied both by the appearance of the Au^0^ absorbance at 240 nm (Figure 6A) and the PR of the GNP (Figure 6B). In contrast, at and above 0.001 M NaCl a peak at 320 nm corresponding to Au^0^ clusters was observed while the PR peak was attenuated. In this case, no PR of GNP was observed even after 24 h of incubation of the reaction mixture at 45 °C suggesting that the GNP formation was completely inhibited. Au^3+^ was reduced but Au^0^ absorbance at 240 nm was low compared to the reaction at 45 °C in the salt-free system (Appendix A). The GNPs were formed at NaCl concentrations below 0.05 M. Like in the case of the lower temperature, at 45 °C the formation of GNP was nearly completely abolished at higher NaCl concentrations (Figure 6B). At all temperatures, when the GNP were formed in the presence of the salt the PR peak was considerably broader and shifted towards higher wavelengths than that in the absence of the salt. Consistent with the shape of the PR peak in the presence of the salt these GNP represented a heterogeneous mixture of larger and smaller particles of different polygonal shapes (Appendix A) compared to much more homogeneous and closer to spherical particles formed without added NaCl (Appendix A). 

### 3.8. Chemical Conversion of the Block Copolymer During the Reaction

Soluble reaction products and GNP were separated by centrifugal filtration, lyophilized, and analyzed by NMR (Figure 7A,B) and IR (Figure 7C). When the reaction was carried out at considerable excess of F127 both the IR and NMR spectra of the separated polymer were nearly identical to the initial F127, albeit the spectra of the purified GNP were markedly different. However, when F127 and HAuCl_4_·3H_2_O were mixed in 1:1 molar ratio (0.25 mM) the spectra of the soluble products were drastically. This suggested that some small amount of F127 was undergoing chemical conversion during the reaction. ^1^H-NMR (Figure 6A) and ^13^C-NMR (Figure 7B) revealed considerable differences between the soluble reaction products and the nanoparticles on the one hand and the initial F127 on the other. Specifically, the ^1^H-NMR revealed that the O–CH_2_ peak at 3.75 ppm was attenuated while several new peaks in the 3.6–3.9 ppm range were observed in both soluble products and nanoparticles. However, while the soluble products displayed the signal of the CH_3_ group of PPO at 1.18 ppm this signal was slightly shifted in the nanoparticles (1.17 ppm) and there was a new peak at 1.30 ppm that was not seen in neither the initial F127 nor soluble products. The differences were even more pronounced in the ^13^C-NMR spectra. Both the soluble products and the nanoparticles displayed new peak at 62 (not observed in initial F127), which can be related to C–OH carbon atoms. However, the nanoparticle spectrum was lacking the peak at 14 ppm (CH_3_) as well as at 73 (O–CCH3) of PPO, suggesting that PPO was not present in GNP. Detailed analysis of IR spectra confirms these observations (Figure 7C). Specifically, in the purified GNP three new IR peaks appeared at 3300, 1030, and 920 cm^−1^ along with the decrease of the peaks in the 1000–1500 cm^−1^ range. This combination of IR spectra changes suggests that C–O bond of F127 was partially cleaved and the OH groups were formed during the reaction. Notably the IR spectra did not support the formation of the carboxylic groups: even though some samples exhibited a minor peak at 1645 cm^−1^ when these samples were carefully dried this peak disappeared completely while the peak at 3300 cm^−1^ remained unchanged. It was previously reported that such behavior can be explained by the presence of water in the copolymer [18]. Furthermore, the CH_3_ bends peak at 1337 cm^−1^ was observed in the initial F127 as well as in soluble products but not in the purified GNP. The degradation of the F127 chain in the process of the reaction was directly demonstrated by GPC (Appendix A). Before the reaction one major peak was seen in the initial 0.5 mM F127 sample with the retention time of 7.03 min, which corresponded to Mw = 9600 Da. The lower value of the molecular mass compared to the listed for F127 copolymer (Mw 12,600 Da), was probably due to compaction of the PPO chains of F127. There was also a peak at 7.59 min (3500 Da), probably corresponding to a diblock copolymer PEO-PPO and/or PPO homopolymer admixtures. After the reaction, the main polymer fraction (7 min) was dramatically decreased. New peaks were observed at 10 min (major), and 12 min (minor), which were below the effective separation range of the GPC column, suggesting that their Mw values were below 700 Da. Analysis of the RI areas under the curve suggested that under the optimized conditions used to produce GNPs 55% of the F127 chains are degraded. Finally, although the GNP did not reveal presence of intact F127, they definitely contained some organic component(s). This was confirmed by TGA analysis demonstrating that the portion of this component was approximately 13% *w*/*w* and it consisted of two compounds with the boiling temperatures of 242 °C and 348 °C (Appendix A).

### 3.9. Batch to Batch Variation in the Formation of GNP

During these studies we tested several different stock solutions of F127 (marked as F127_1, F127_2 and F127_3). The polymers in these solutions had essentially the same chemical structure and molecular weight as confirmed by IR, NMR, and GPC of lyophilized aliquots. It is important to note that during lyophilization volatile molar mass degradation products, which might have been present in the solution, were lost and could not be detected by any of the applied methods. However, one difference between these batches was in the extents of polymer oxidation (Appendix A). Specifically, the hydroperoxides level in F127_3 was 2–3 times higher than these in F127_1 and F127_2 (Appendix A). The free radical levels were comparable in all three solutions. Comparison of these batches discovered considerable batch-to-batch variation in the formation of GNP. Thus, reacting HAuCl_4_·3H_2_O with F127_1 resulted in the GNP with UV characteristics and sizes similar to these of the GNP described above. However, reacting HAuCl_4_ with F127_2 resulted in a relatively broader PR peak centered at 600 nm and a larger hydrodynamic size (156 nm) of purified GNP (Figure 8A, Table 4). In the case of F127_3, the UV spectra of the reaction mixture exhibited 1) a shift in the absorbance from 220 to 240 nm indicative of Au^3+^ reduction to Au^0^; and 2) appearance of a peak at 330 nm, corresponding to formation of Au^0^ clusters (Figure 8A). However, no PR peak was observed in this case, suggesting that GNP did not form. Indeed, albeit DLS of unpurified reaction mixture showed presence of 500 nm aggregates (Appendix A), no GNPs were isolated after standard purification procedure (Table 4). Among notable differences between the batches were the larger particle sizes (60 nm) and more acidic pH of the F127_3 stock solution (pH 5.1) compared to the solutions of two other polymer samples (pH 6.8–6.9) (Appendix A). Since we previously established that the formation of the GNP is dependent on the pH of the copolymer solution, we adjusted the pH of the F127_3 solution to either weakly acidic pH 6.5 or alkaline pH 11.6. Notably, pH adjustment did not decrease the particle size (74 nm at pH 6.5 and 81 nm at pH 11.6), which remained much larger than normally expected for F127 micelles (27 nm) (Appendix A). However, reacting HAuCl_4_·3H_2_O with the F127_3 solutions with adjusted pH proceeded differently and, in both cases, resulted in formation of GNP. Thus, UV spectra suggested that after mixing F127_3 solutions, pH 6.5 with HAuCl_4_·3H_2_O the reduction of Au^3+^ was decelerated as was evident by the presence of a large peak of Au^3+^ at 220 nm (Figure 8A). At the same time, the peak of the Au^0^ clusters at 330 nm disappeared. However, there was a small and broad PR peak, suggesting formation of GNPs. The DLS has shown a presence of 280 nm aggregates in the reaction mixture (Appendix A), which after purification produced 107 nm GNPs (Table 4). When the reaction was carried out using the alkali solution of F127_3, pH 11.6 the reduction of Au^3+^ was complete, and the large PR peak of GNP was observed. Overall, the UV spectra were similar to these observed after conducting the reaction under basic conditions (Figure 4E,F). Moreover, the particle sizes were also nearly identical (compare Table 2 and Table 3). 

### 3.10. Effect of Lyophilization of the Copolymer Solutions on GNP Formation

In an attempt to remove hydroperoxides the polymer solutions were lyophilized and re-dissolved in the same volume of bidistilled water using a previously described protocol [19]. There was little change (less than 15%) in the hydroperoxides in F127_1, approximately 2-fold increase in the hydroperoxides in F127_2, and nearly 6-fold decrease in the hydroperoxides in F127_3 (Appendix A). Moreover, lyophilization had little if any effect on the free radical levels, which were similar in all studied samples. Although there was no effect on the pH in F127_1 and F127_2 solutions, the pH of the reconstituted F127_3 solution increased from pH 5.1 to pH 6.7 (Appendix A). At the same time the effects of lyophilization of F127_3 samples on the Au^3+^ reduction and GNP formation were distinct and profound. Specifically, in contrast to F127_3 solution before lyophilization the lyophilized and reconstituted F127_3 solution was able to reduce Au^3+^ and promote formation of GNP when mixed with HAuCl_4_ (Figure 8B and Table 4).

### 3.11. Formation of Reacting Oxygen Species during the Reaction

This study measured changes in the content of free radicals and hydroperoxides before and after mixing of F127 solution with HAuCl_4_·3H_2_O (Figure 9A). At 2 h the content of free radicals remained relatively small, however, there was a very considerable increase in the hydroperoxides. To elucidate the role of oxygen in the formation of the hydroperoxides in the course of the reaction we purged the copolymer solution with nitrogen for 5 min before initiating the reaction. This resulted in attenuation of formation of the hydroperoxides in the reaction mixture (Figure 9A). In this case, we did not register the UV spectra to avoid exposure of the solutions to the atmospheric oxygen. However, the visual observation of the reaction mixtures at 2 h revealed that the color of the solutions was different—pink in before purging (Figure 9B) and purple after purging (Figure 9C). This suggested that the PR has red-shifted after purging, which is indicative of the alteration of the particle size and shape.

### 3.12. Mechanism of HAuCl_4_·3H_2_O Reduction by Pluronic

Analysis of the data allows to propose the following schematic for HAuCl_4_·3H_2_O reduction in the presence of Pluronic:The decomposition of the hydroperoxides present in Pluronic leads to Pluronic chains degradation and formation of lower molecular mass alcohols. This process results in the release of a hydroxonium ion that contributes to acidification of the media and formation of the superoxide O_2_^−^^•^ and hydroxyl radical HO^•^ or hydrogen peroxide that can participate in further redox reactions. Here we present the oxidation and decomposition of the PPO block because it is known to degrade faster than the PEO [20]. However, similar processes can also proceed at the PEO block and PEG homopolymer.

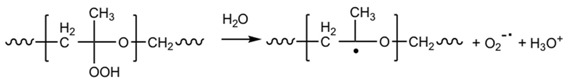
(1)

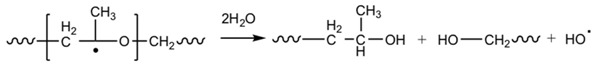
(2)
2HO^•^ → H_2_O_2_(3)The reactive oxygen species formed in (1), (2), and (3) promote the reduction of Au^3+^. Since tetracloroauric ion [AuCl_4_]^−^ undergoes acid and alkalli hydrolysis and gradually transforms into tetrahydroxoaurate ion [Au(OH)_4_]^−^ at basic conditions [21] the reduction reactions proceed through different mechanisms depending on pH. Most likely these reactions are complex and involve multiple elementary steps and intermediates. Their formal description is further complicated by the coexistence of several forms of aquachlorohydroxo complexes of gold(III) that can participate in the reducton with varying reactivity. Therefore, we present only the simplified schemes of the overall reactions with either tetracloroauric or tetrahydroxoaurate ions at acidic or alkaline conditions, respectively. We also would like to point out that the reactivities of the superoxide radical O_2_^−^^•^ and hyrogen peroxide can depend on pH. The superoxide radical at low pH exists mainly in the protonated form of the hydroperoxyl radical HO_2_^•^ (pKa ~4.8), while the hyrogen peroxide (pKa ~11.6) at alkaline conditions can fom the hydroperoxide anion HO_2_^−^.In acidic conditions
O_2_^−•^ + H_3_O^+^ → HO_2_^•^ + H_2_O(4)
[AuCl_4_]^−^ + 3HO_2_^•^ + 3H_2_O → Au^0^ + 3O_2_ + 3H_3_O^+^ + 4Cl^−^(5)
[AuCl_4_]^−^ + 3O_2_^−•^ → Au^0^ + 3O_2_ + 4Cl^−^(6)
2[AuCl_4_]^−^ + 3H_2_O_2_ + 6H_2_O → 2Au^0^ + 3O_2_ + 6H_3_O^+^ + 8Cl^−^(7)In alkaline conditions
Au(OH)_4_^−^ + 3O_2_^−•^ → Au^0^ + 3O_2_ + 4OH^−^(8)
H_2_O_2_ + OH^−^ → H_2_O + HO_2_^−^(9)
2Au(OH)_4_^−^ + 3HO_2_^−^ → 2Au^0^ + 3O_2_ + 5OH^−^ + 3H_2_O(10)Reactions (5), (6), (7), (8), and (10) are reversible, but we assume that their equilibria are shifted to the right due to the rapid consumption of the molecular oxygen in subsequent reactions with Pluronic. The latter may be accelerated by the complexation of [AuCl_4_]^−^ and related gold complexes with the block copolymer chains, as in this case the polymer oxidation may proceed either concurrently or immediately after the reduction of the gold ions.Finally, the molecular oxygen formed in the reduction reactions (5), (6), (7), (8), and (10) propagates hydroxyperoxidation of either PPO or PEO chains in Pluronic.

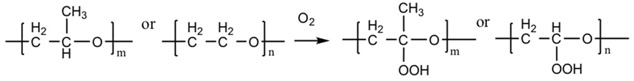
(11)

## 4. Discussion

Previous studies have demonstrated formation of GNPs in the presence of Pluronics without use of any standard reducing agent such as sodium citrate or NaBH_4_ [22]. The mechanism postulated in these works includes: 1) reduction of [AuCl_4_]^−^ ions by PEO chains leading to the formation of Au^0^ clusters; 2) adsorption of the PPO chains of the block copolymers on these clusters with subsequent reduction [AuCl_4_]^−^ on the cluster surface; and 3) growth of the GNPs stabilized by the adsorbed block copolymers [9]. In this mechanism known as ‘three-step reduction and particles formation’ the first two stages correspond to the nucleation of GNPs, and the third stage to the growth or the nuclei into mature nanoparticles. The mechanism suggests that the copolymer plays a crucial role in both processes. However, to the best of our knowledge, the specific steps in the mechanism and the resulting products have not been not precisely defined and/or separated from the reaction mixture. The current GNPs formation theory suggests that the first stage of the process involves formation of the ‘crown ether-like structures’ between [AuCl_4_]^−^ ions and PEO chains followed by reduction of Au^3+^ and oxidation of PEO. This is based on an early study published by Longenberger and Mills on HAuCl_4_·3H_2_O reduction by PEO homopolymer [6]. They reported that the formation of GNPs was accelerated, and the particles became more stable as the PEO molar mass increased. Therefore, it was proposed that, for full reduction, Au^3+^ ion should migrate through several ether crown structures. Further work suggested that metal ions interact with the hydrated PEO blocks of the block copolymers but are immiscible with their PPO blocks [23]. Thus it was suggested that PPO blocks cannot directly reduce metal ions but are essential for well-defined shape controlled synthesis of gold nanostructures in mesophase media with low water content [23]. Sakai and Alexandridis compared PEO homopolymer to various Pluronics and concluded that the reaction activity of the latter is much higher than that of the PEO [7,9]. While these authors noted that the PPO block may be sufficiently hydrated to interact with metal ions, they stopped short of suggesting that PPO can actually participate as a reagent in the reduction reaction. However, they suggested that PPO blocks play significant role in the later stages of the process by absorbing at the particle surface and limiting the particle growth. 

In this work, we separated the GNPs and soluble polymer products and analyzed them by several physicochemical methods. The results clearly suggested that the copolymer undergoes chemical decomposition in the process of the reaction including cleavage of the C–O bonds (IR), depletion of the O–CH_2_ signal (^1^H-NMR), and decrease of the molecular mass of the copolymer (GPC). Although the purified nanoparticles contained about 13% by weight of the organic component (by TGA), we did not find any presence of PPO groups in them suggesting that PPO chains or intact Pluronic molecules are not adsorbed on the particles surface. Interestingly, the analysis of soluble products did reveal presence of the CH_3_ group of PPO. The product analysis also indicated presence of newly formed OH groups both in the purified GNPs and in the soluble products (IR), which according to ^13^C-NMR correspond to primary hydroxyl C–(OH) groups. In contrast to previous reports suggesting that oxidation of the PEO proceeds all the way to the carboxylate groups [6], IR, ^1^H-NMR, or ^13^C-NMR did not confirm the presence of these groups in the purified GNPs or reaction products. Hence, the purified GNPs appear to contain lower molecular mass alcohols produced as a result of Pluronic degradation that are likely attached to the nanoparticle surface rather than embedded within their volume. A very interesting phenomenon was observed when the reaction was carried out at the surface of F127 gel, with the gel being dissolved in the course of the frontal reaction. This suggests that cleavage of Pluronic chains during the reaction can disrupt self-assembly of the block copolymer aggregates (in the specific case—the copolymer gel).

We also discovered that the reaction of the block copolymer and HAuCl_4_ involves formation of free radicals and hydroperoxides as their concentration is sharply increased. Interestingly, we found that the amount of the hydroperoxides decreases in the solutions purged with nitrogen and therefore appears to be dependent on the concentration of the oxygen. At the same time the nitrogen-purged solutions reveal some increase in the free radicals suggesting that the oxidation mechanisms are complex and involve multiple intermediates. It is noteworthy that the reaction proceeding under the nitrogen atmosphere resulted in the formation GNPs with different spectral characteristics than the reaction carried out under the air. Moreover, we found that F127 batches having different levels of the free radicals and hydroperoxides behave differently in the reaction. This is an important result for all such studies suggesting that the underlying processes are affected by the initial degree of oxidation of the polymers. These processes are hard to control and at the same time they can contribute to Pluronic batch-to-batch variation.

Taken together, we propose a multi-stage process for the reduction of HAuCl_4_·3H_2_O by Pluronic, see reactions (1)–(11). It is well known that polyethers including poloxamers can undergo oxidation and formation of the free radicals and hydroperoxides [19]. Hence, we posit that at the first, initiation stage the Pluronic hydroperoxides decompose in reactions (1) and (2) resulting in polymer degradation to lower molecular mass primary alcohols and formation of the reactive oxygen species (1–3), that drive the subsequent reduction of various gold(III) forms. In the process of polymer degradation the hydronium ions are also released, which explains a pH decrease (by about 2 pH units) observed experimentally. At the second stage, gold(III) is reduced to Au^0^ in complex processes (4–10) involving multiple elementary steps also leading to formation of molecular oxygen. Some of these reactions are probably similar to previously reported reactions of reduction of transition metals by the reactive oxygen species [24]. These processes are pH dependent due to the presence of several aquachlorohydroxo complexes of gold(III) and various forms of reactive oxygen species, O_2_^−^^•^, HO_2_^•^, H_2_O_2_, and HO_2_^−^, with different reactivity [25]. The pH dependence of the reaction was observed experimentally in this study as well as in previous work that reported that formation of GNPs in the presence Pluronic P123 was inhibited at acidic conditions (≈pH 2.0) and could be reversed by pH adjustment [10]. Finally, at the third stage, the molecular oxygen formed during the gold(III) reduction propels further oxidation of Pluronic to hydroperoxides (11) and propagation of the block copolymer degradation. This explains a burst in the hydroperoxides observed experimentally. It is possible that reaction (11) proceeds in gold-Pluronic complexes either simultaneously (intramolecularly) or right after elementary steps of the gold ion oxidation and the O_2_ formation.

The effect of the Pluronic self-assembly on the formation of GNPs observed in this study is also of considerable interest. Even though Pluronic molecules do not appear to strongly bind with the GNPs (as they can be separated by centrifugal filtration), the GNPs formation kinetics and final properties are strongly affected by the block copolymer self-assembly. First, there was a clear dependence of the shape of GNPs on Pluronic concentration. As the concentration increased the particles became more homogeneous and spherical. Second, at low temperature when the self-assembly of Pluronic was suppressed the formation of GNPs was impaired although the reduction of Au^3+^ proceeded. Hence, we believe that Pluronic aggregates serve as templates for the GNPs growth. Finally, formation of aggregates that are larger than both individual GNPs and Pluronic micelles was observed. These aggregates represented clusters of GNPs grains interconnected by the block copolymer and they disintegrated upon dilution. Finally, our study suggests that F127 having the longest PEO and PPO chains forms the most stable nanoparticles. Colloidal stabilization of pre-formed GNPs by Pluronics has been previously reported [26]. The stabilization effect enhanced as the PPO or PEO chain length increased. This behavior was attributed to the formation of the polymer layers surrounding the GNPs rather than adsorption of PPO chains directly at the GNPs surface.

Finally, of interest are the effects of various reaction conditions on the shape of the GNPs. It is known that the rate of production of the reduced metal precursors—i.e., Au^0^—can influence the nanoparticle shape. Rapid production of Au^0^ facilitates formation of thermodynamically favored polyhedral (spherical) GNPs, while slower production of Au^0^ usually results in heterogeneous GNPs shapes. For example, reacting HAuCl_4_ with a strong reducing agent such as NaBH_4_ can lead to a rapid formation of the Au^0^ nuclei followed by the fast growth of spherical GNPs [27,28]. In contrast, when a mild reductant, such as phenylenediamine, PVP or glucose, is used the GNPs growth is kinetically controlled resulting in the formation of hexagonal and triangular particles. This consideration helps to explain the effect of Pluronic F127 concentration on the GNPs shape observed in this study. As the concentration of the copolymer increases Au^0^ production becomes faster and the particles become more uniform and homogeneous. The effect of HAuCl_4_·3H_2_O concentration appears to be more complex. We observed formation of small spheres at low concentrations of HAuCl_4_·3H_2_O, followed by larger spheres, multiple shapes, and spheres again as the HAuCl_4_·3H_2_O concentration increased. Since HAuCl_4_·3H_2_O contained some amount of Au^0^ upon change of its concentration there was likely a change of both the nuclei and metal precursor. Another factor, which affects the GNPs shape, is pH. In the present study at low pH we observed slow formation of heterogeneous multiform GNPs, which is explained by slower Au^0^ production due to a decrease of the reduction potential of the reactive oxygen species [25]. As the pH increased the particles became more homogeneous, spherical, smaller, and stable. Prior studies concluded that pH had similar effect on the size and shape of GNPs regardless of the type of the reductant, such as Tetronic T904 [29], Pluronics [10], or polyols [30]. Moreover, similar effects of pH were observed for silver [31] and platinum nanoparticles [32]. Previously, chloride ions were reported to promote formation of triangular GNPs [33]. We also observed that addition of small amounts of NaCl to the reaction medium results in the formation of heterogeneous, polygonal GNPs. Surprisingly however, as the NaCl concentration further increased, there was complete inhibition of the GNP formation. This phenomenon may be explained by the effects NaCl on the interactions of the block copolymer with Au^0^ clusters or GNPs, since NaCl is known to promote dehydration of PEO and PPO chains, enhance crystallization of PEO chains, and affect the micelle structure [34,35]. Therefore, by varying the concentration of NaCl in the system, one can modify the process parameters and the shape of the formed particles.

## 5. Conclusions

In conclusion, this work presents a detailed study on the formation of the GNP as a result of reduction of chloroauric acid with Pluronic block copolymers. We demonstrated that in the course of this reaction the copolymer chains undergo degradation and become enriched with primary hydroxyl groups, although the oxidation of the polymer does not go all the way down to the carboxylic acids. The purified nanoparticles also contain about 13% by weight of the organic component but can be fully separated from the excess of the copolymer. The reaction involves formation of free radicals and hydroperoxides and appears to be dependent on the concentration of the oxygen in the reaction mixture. We elucidated various environmental factors such as pH, temperature, the sodium chloride concentration, as well as the concentrations of the reactants that affect the rate of the reaction and the yield and morphology of the resulting GNP. The findings advance understanding of the reaction mechanisms and could be useful for improving the methods of manufacturing uniform and pure nanoparticles for various technical applications including nanomedicine, where GNPs could be used in numerous medical applications, from diagnostics to therapy [36,37].

## Figures and Tables

**Figure 1 polymers-11-01553-f001:**
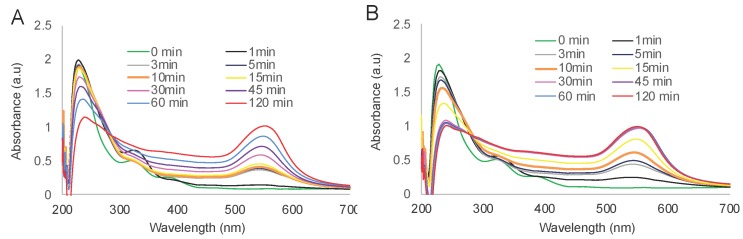
The time evolution of the UV–Vis spectra in the mixtures containing 6.3% *w*/*w* F127 and 0.25 mM HAuCl_4_ during incubation at 25 °C (**A**) or 45 °C (**B**).

**Figure 2 polymers-11-01553-f002:**
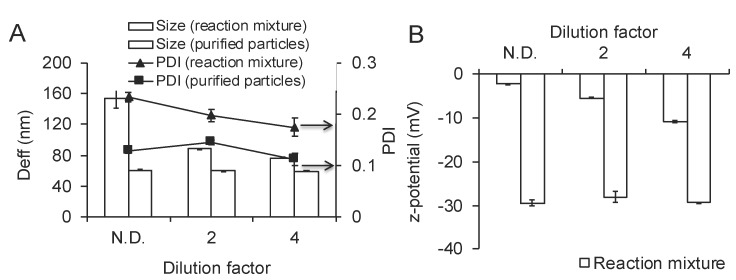
Changes in the (**A**) particle effective diameter (columns) and PDI (lines) and (**B**) ζ-potential potential following dilution of reaction mixture and purification of GNPs.

**Figure 3 polymers-11-01553-f003:**
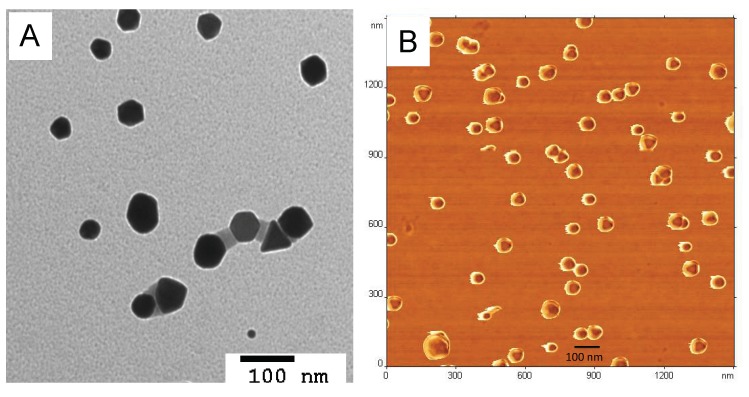
(**A**) TEM and (**B**) AFM images of purified gold nanoparticles.

**Figure 4 polymers-11-01553-f004:**
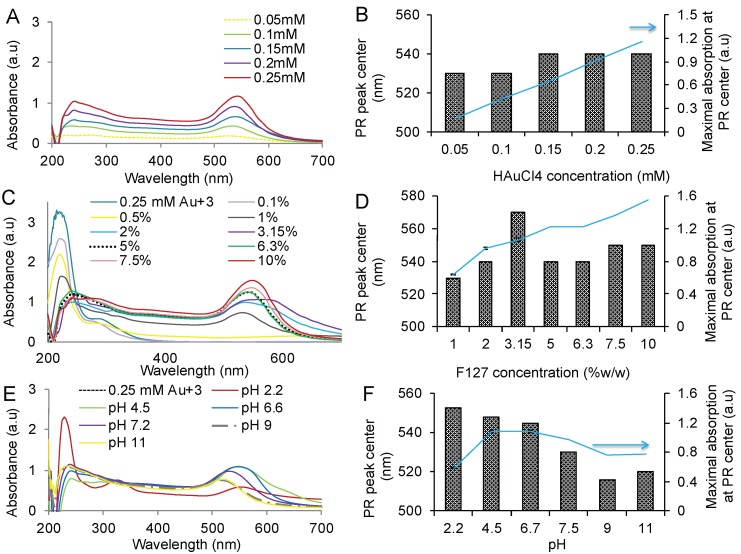
(**A**,**C**,**E**) Absorbance spectra, and (**B**,**D**,**F**) the position of the PR peak maxima (columns) and the maximal PR absorbance intensity (▴) in the mixtures containing HAuCl_4_ and F127 after 2 h incubation at 45 °C. (**A**,**B**) HAuCl_4_ concentration is constant (0.25 mM) and F127 concentration varied from 0.1 to 10 % *w*/*w*. The pH of the F127 solution before HAuCl_4_ addition was pH 6.5. In all cases reaction is complete based on the absence of a peak at 220 nm or shift of the pick to 240 nm for 0.25 mM. (**C**,**D**) F127 concentration is constant (6.3% *w*/*w*), HAuCl_4_ concentration is varied from 0.05 to 0.25 mM. The pH of the F127 solution before HAuCl_4_ addition was pH 6.5. (**E**,**F**) 0.25 mM HAuCl_4_ and 6.3% *w*/*w* F127 mixture was adjusted to various pH.

**Figure 5 polymers-11-01553-f005:**
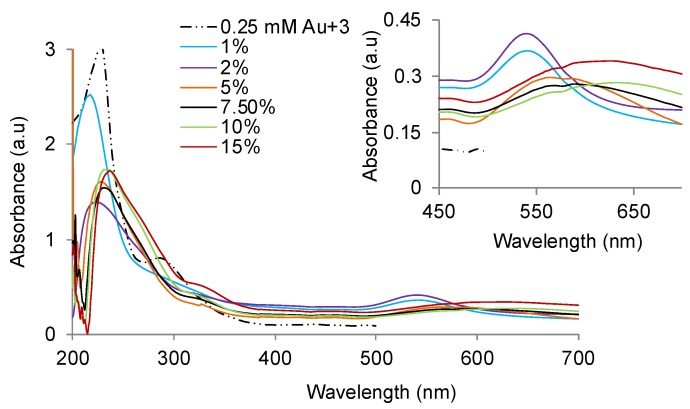
UV–Vis absorbance spectra and (insert) absorbance in PR region in the mixtures of 0.25 mM HAuCl_4_ and F127 as a function of the copolymer concentration 2 h after incubation at 4 °C. Copolymer concentration varied from 1–15% *w*/*w*.

**Figure 6 polymers-11-01553-f006:**
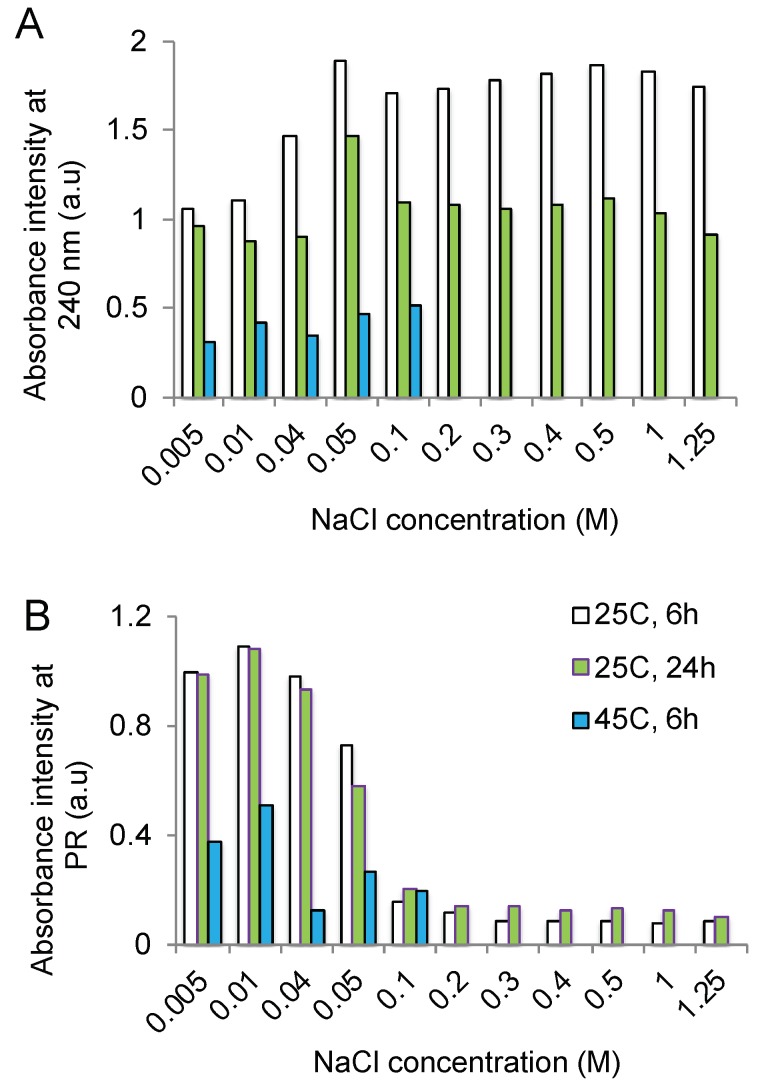
(**A**) Absorbance intensity at 240 nm and (**B**) the maximal absorbance intensity of the PR peak in the mixtures of 0.25 mM HAuCl_4_ and 6.3% *w*/*w* F127 as a function of the NaCl concentration after incubation of the reaction mixture at 25 °C or 45 °C at 6 h and 24 h.

**Figure 7 polymers-11-01553-f007:**
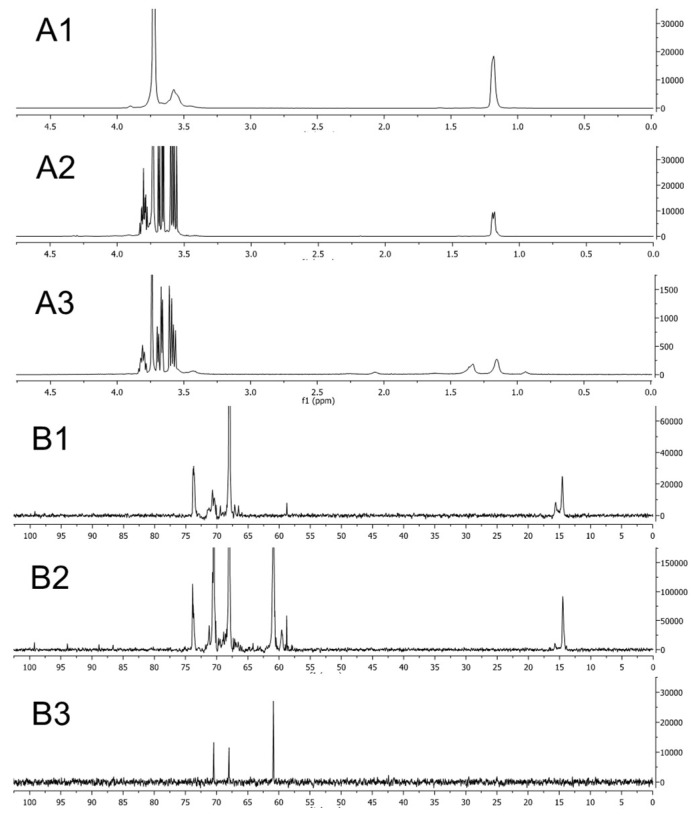
(**A**) ^1^H-NMR, (**B**) ^13^C-NMR, and (**C**) the IR spectra of (A1, B1, C1) original F127, (A2, B2, C2) separated reaction mixture and (A3, B3, C3) the purified gold nanoparticles.

**Figure 8 polymers-11-01553-f008:**
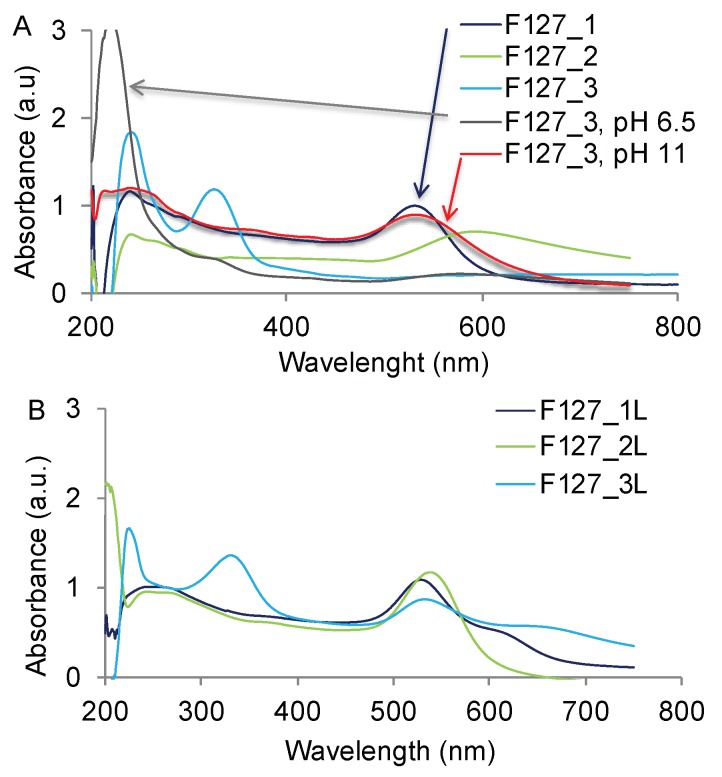
Absorbance spectra in the mixtures of 0.25 mM HAuCl_4_ and different F127 batches (**A**) before and (**B**) after lyophilization. The HAuCl_4_ concentration was 0.25 mM and F127 concentration was 6.3%. The reaction mixtures were incubated for 2 h (unless shown 1 h) at 45 °C.

**Figure 9 polymers-11-01553-f009:**
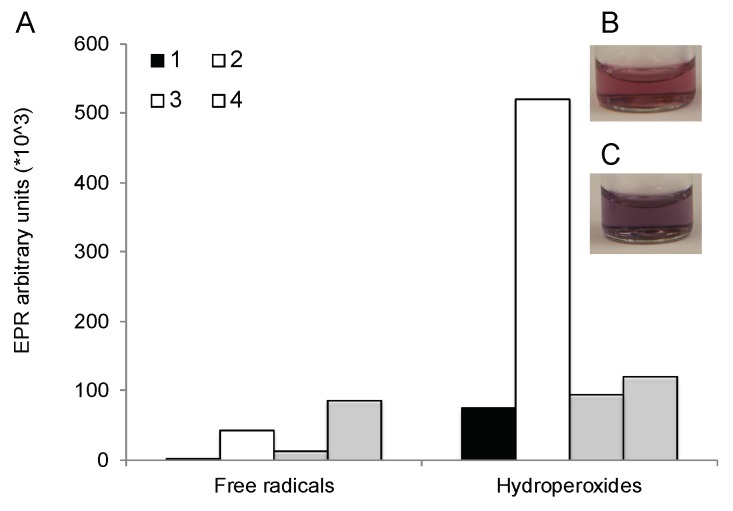
(**A**) Free radicals and hydroperoxides content in the regular (1,2) and N_2_ purged (3,4) stock solutions of 6.3% *w*/*w* F127 (1,3) and the reaction mixtures of these solutions with HAuCl_4_ (2,4). (**B**,**C**) Images of GNP dispersions formed in (**B**) regular and (**C**) N_2_ purged solutions of 6.3% *w*/*w* (5 mM) F127. The F127 solution was purged with N_2_ for 5 min before mixing with HAuCl4, and then the reaction was carried in a closed vessel. In all cases, the reaction mixtures were incubated for 2 h (unless shown 1 h) at 45 °C.

**Table 1 polymers-11-01553-t001:** Effect of HAuCl_4_ concentration on the characteristics of the GNP ^1^.

HAuCl_4_ Concentration (mM)	DLS (Reaction Mixture)			DLS (Purified Particles)			Morphology by TEM
	D_eff_ (nm)	PDI	ζ-Potential (mV)	D_eff_ (nm)	PDI	ζ-Potential (mV)	
0.05	50 ± 1	0.53 ± 0.02	−3.76 ± 0.11	77 ± 16	0.18 ± 0.04	−30.23 ± 0.05	Multi-sized spheres
0.10	122 ± 1	0.34 ± 0.001	−3.52 ± 0.03	62 ± 1	0.25 ± 0.01	−28.13 ± 1.29	Multi-sized spheres
0.15	216 ± 1	0.23 ± 0.01	−3.18 ± 0.01	79 ± 1	0.19 ± 0.01	−28.70 ± 3.41	Triangles, rods, spheres
0.20	231 ± 1	0.22 ± 0.02	−2.45 ± 0.02	87 ± 1	0.14 ± 0.03	−27.47± 0.06	Triangles, plates, spheres
0.25	229 ± 2	0.23 ± 0.01	−3.01 ± 0.02	72 ± 1	0.12 ± 0.02	−37.73 ± 0.25	Spheres, triangles

^1^ Particles were synthesized by mixing equal volumes of F127 solution (final concentration 6.3 % *w*/*w*, pH 6.5) and HAuCl_4_ solutions of different concentrations, followed by incubation of the mixture for 2 h at 55 °C. Particles were purified from excess of the polymer by centrifugal filtration at 1500 rpm for 30 min at 4 °C (repeated three times). The purified nanoparticles were restored to original volume with distilled water (particles concentration remained the same before and after purification).

**Table 2 polymers-11-01553-t002:** Effect of F127 concentration on the characteristics of the GNP ^1^.

F127 Concentration (% *w*/*w*)	DLS (Reaction Mixture)			DLS (Purified Particles)			Morphology by TEM
	D_eff_ (nm)	PDI	ζ-Potential (mV)	D_eff_ (nm)	PDI	ζ-Potential (mV)	
1.0	121 ± 1	0.28 ± 0.01	−12.13 ± 0.30	80 ± 1	0.29± 0.01	−25.57 ± 0.40	Multi-sized spheres, rods, hexagons
2.0	91 ± 1	0.43 ± 0.01	−6.45 ± 0.30	54 ± 1	0.42 ± 0.03	−30.20 ± 0.66	Multi-sized spheres, rods, hexagons, triangles
3.15	101 ± 2	0.28 ± 0.01	−5.52 ± 0.02	62 ± 1	0.29 ± 0.003	−28.20 ± 0.20	Spheres, triangles, rods
5.0	133 ± 2	0.22 ± 0.01	−4.04 ± 0.20	63 ± 1	0.19 ± 0.02	−32.23 ± 1.85	Spheres, triangles
6.3	153 ± 12	0.23 ± 0.01	−3.84 ± 0.06	58 ± 1	0.17 ± 0.01	−36.50 ± 0.38	Spheres, triangles
7.5	227 ± 6	0.17 ± 0.02	−3.76 ± 0.20	75 ± 1	0.15 ± 0.01	−41.80 ± 0.56	Spheres
10.0	305 ± 3	0.11 ± 0.02	−2.83 ± 0.30	86 ± 1	0.09 ± 0.01	−45.30 ± 0.15	Uniform spheres

^1^ Particles were synthesized by mixing equal volumes of F127 and HAuCl_4_ solutions (final concentrations 1–10% *w*/*w*, pH 6.5, and 0.25 mM respectively) followed by incubation of the mixture for 2 h at 55 °C. Particles were purified as described in Table 1. The purified nanoparticles were restored to original volume with distilled water (particles concentration remained the same before and after purification).

**Table 3 polymers-11-01553-t003:** Effect of pH of F127 solution on the GNP characteristics ^1^.

pH of the F127 Solution ^a^	DLS (Reaction Mixture)			DLS (Purified Particles)			TEM (Purified Particles)	
	D_eff_ (nm)	PDI	ζ-Potential (mV)	D_eff_ (nm)	PDI	ζ-Potential (mV)	D_eff_ (nm)	Morpho-Logy
2.2 (1.24) ^2^	375 ± 3	0.11 ± 0.3	−2.7 ± 0.05	114 ± 1	0.17 ± 0.02	−25.8 ± 3.13	n.d.	Spheres, rods, triangles
4.5 (1.44) ^2^	219 ± 3	0.20 ± 0.24	−3.07 ± 0.13	85 ± 1	0.16 ± 0.01	−32.53 ± 0.58	77 ± 6	Spheres
6.7 (3.14) ^3^	160 ±2	0.19 ± 0.04	−3.98 ± 0.18	72 ± 1	0.12 ± 0.02	−37.70 ± 0.25	54 ± 13	Spheres
7.5 (5.5) ^4^	115 ±3	0.36 ± 0.04	−4.50 ± 0.66	57 ± 1	0.26 ± 0.01	−33.77 ± 0.66	23 ± 11	Spheres
9.0 (6.35) ^4^	56 ±1	0.56 ± 0.01	−4.45 ± 0.21	53 ± 1	0.36 ± 0.02	−35.40 ± 1.37	12 ± 6	Spheres
11.5 (7.76) ^4^	65 ± 4	0.44 ± 0.02	−5.20 ± 0.62	41 ± 1	0.36 ± 0.03	−32.06 ± 1.45	12 ± 4	Spheres

Particles were synthesized by mixing equal volumes of F127 and HAuCl_4_ solutions (final concentrations 6.3% *w*/*w* and 0.25 mM, respectively) followed by incubation of the mixture for 2 h at 45 °C. ^1^ The pH values of the initial 12.6% Pluronic solution and final reaction mixture at 2 h (in the brackets) are presented. The pH in the Pluronic solution (pH 6.7) was adjusted with ^2^ 1N HCl, ^3^ not adjusted, ^4^ 1N NaOH. After mixing this solution with HAuCl_4_ the pH value rapidly (mins) decreased and then did not change in the course of the reaction. n.d.—not determined.

**Table 4 polymers-11-01553-t004:** Characteristics of purified GNP synthesized using non-lyophilized and lyophilized F127 samples ^1^.

F127 Batch	GNP Synthesized Using Non-Lyophilized F127		GNP Synthesized Using Lyophilized F127	
	D_eff_ (nm)	PDI	D_eff_ (nm)	PDI
F127_1	62 ± 1	0.16 ± 0.01	47 ± 1	0.30 ± 0.004
F127_2	157 ± 2	0.06 ± 0.02	70 ± 1	0.07 ± 0.01
F127_3	Not formed	Not formed	57 ± 1	0.30 ± 0.04
F127_3 (pH 6.5)	108 ± 2	0.17 ± 0.003	n/a	n/a
F127_3 (pH 11.6)	39 ± 3	0.51 ± 0.12	n/a	n/a

^1^ The 12.6% stock solutions of F127 solutions were prepared by dissolving the copolymer in bidistilled water. These solutions were lyophilized and restored to the same volume by bidistilled water. GNP were synthesized by mixing equal volumes of non-lyophilized and lyophilized F127 solutions and HAuCl_4_ solution (final concentrations 6.3% *w*/*w* F127 and 0.25 mM HAuCl_4_) followed by incubation of the mixture for 2 h at 45 °C. Particles were purified as described in Table 1. The purified GNPs were restored to the original volume using bidistilled water (particles concentration remained the same before and after purification).

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
