# Peer review of "Synthesis of Well-Defined Gold Nanoparticles Using Pluronic: The Role of Radicals and Surfactants in Nanoparticles Formation"

_polymers, 2019, doi:10.3390/polym11101553_

Round 1

Reviewer 1 Report

The authors report on the study of the reduction of HAuCl4 in water in the presence of PEO-PPO-PEO block copolymers (Pluronics). Several analysis techniques are used to assess the reaction products as a function of the reactants concentrations and reaction conditions (pH, salt, temperature). As opposed to the authors' claim, this work does not appear to provide any insight into the mechanism of HAuCl4 reduction and GNP formation in the presence of Pluronic block copolymers, and the low relevance of the reported results and supported conclusions is not commensurate with the length of their detailed and rather confuse description. The discussion section is more a review of previous literature, than a discussion of the reported results. I therefore do not recommend publication of this manuscript. More specific comments are given below.

The initial mention of several copolymers is useless since only one is being studied. Produced GNPs are described as stable in colloidal suspensions, but some aggregation is mentioned to explain the discrepancy between DLS and TEM determinations of their size. They are described as uniform and well-defined, while TEM images, provided in the supporting information, show that the uniformity is far from the state of the art and from what is required for nanomedicine applications.

The experimental kinetic data (for instance figure 2, which would probably be more readable if displayed as the time evolution of the UV-vis spectra) does not clearly support the suggestion of a nucleation/growth two-step process.The low temperature experiment is little convincing since low temperature can hinder any redox reaction regardless of environmental factors (here te presence of Pluronic micelles).

Size values given to the tenth of nm (61.6 ± 0.4 nm) do not make sense in view of the TEM images (S3 and S4 for instance). Such images would be more readable if tiled at identical scale.

Refs 19 and 31 are incomplete.

Author Response

Reviewer 1

The authors report on the study of the reduction of HAuCl4 in water in the presence of PEO-PPO-PEO block copolymers (Pluronics). Several analysis techniques are used to assess the reaction products as a function of the reactants concentrations and reaction conditions (pH, salt, temperature). As opposed to the authors' claim, this work does not appear to provide any insight into the mechanism of HAuCl4 reduction and GNP formation in the presence of Pluronic block copolymers, and the low relevance of the reported results and supported conclusions is not commensurate with the length of their detailed and rather confuse description. The discussion section is more a review of previous literature, than a discussion of the reported results. I therefore do not recommend publication of this manuscript. More specific comments are given below.

We respectfully disagree with the critique. First, the manuscript is providing systematic data on the effects of environmental conditions of the reaction of synthesis of gold nanoparticles in the presence of Pluronic block copolymers that substantially extend previous literature. Second, we for the first time demonstrate that the reaction of the synthesis of the gold nanoparticles proceeds with the degradation of Pluronic chains such as PPO and analyze the products of the reaction. Previous literature did not uncover this, because most authors did not purify the gold nanoparticles and did not separate the products of degradation after the synthesis and/or the reactions were carried out in considerable excess of Pluronic that masked the degradation products. Finally, we demonstrate a drastic increase of the generation of hydroperoxides and superoxides in the course of the reaction, that to the best of our knowledge was not shown before.  Based on these findings clearly demonstrate that prior studies declaring the mechanism of the reaction are incomplete or not correct. In response suggestion of the reviewer 2 we now added the mechanistic scheme of the reaction processes that are consistent with our data (Section “3.13 Mechanism of HAuCl4•3H2O reduction by Pluronic.”)

We appreciate the critical comment regarding the length of the description. We have clarified and shortened the discussion by nearly two pages (two-fold). In conclusion, while we disagreed with the critique we believe that it has been very helpful and allowed us to significantly improve the paper.

We have shortened the manuscript when advisable.

The initial mention of several copolymers is useless since only one is being studied.

Thank you. Figure 1 and preliminary considerations for the choice of a lead copolymer were moved to the supplementary materials.

Produced GNPs are described as stable in colloidal suspensions, but some aggregation is mentioned to explain the discrepancy between DLS and TEM determinations of their size. They are described as uniform and well-defined, while TEM images, provided in the supporting information, show that the uniformity is far from the state of the art and from what is required for nanomedicine applications.

First, we demonstrate that not all GNPs batches synthesized using Pluronic under different conditions are uniform and well-defined. It appears that the reviewer refers to the results of the synthesis under acidic conditions – where we emphasize the lack of uniformity of the gold nanoparticles both by DLS and TEM. Second, we demonstrate and describe the conditions under which the uniform and well-defined GNPs are formed (e.g., GNPs prepared by mixing equal volumes of F127 solution, 10% w/w, pH 6.5 and 0.25 mM HAuCl4•3H2O, followed by incubation of the mixture for 2h at 55oC, see Table 2). The uniformity is described in terms of the DLS polydispersity index (PDI), which is standard for nanomedicine. The PDI values for optimal synthesis conditions are around 0.01 and for those conditions the uniformity is confirmed by TEM. This value of polydispersity is state of the art and excellent for nanomedicine use. Therefore, the implication of our study is that state of the art, uniform gold nanoparticles with PDI within the required range can be produced using only if one selects the optimal synthesis conditions. This is both significant and new.

The experimental kinetic data (for instance figure 2, which would probably be more readable if displayed as the time evolution of the UV-vis spectra) does not clearly support the suggestion of a nucleation/growth two-step process. The low temperature experiment is little convincing since low temperature can hinder any redox reaction regardless of environmental factors (here the presence of Pluronic micelles).

The nucleation/growth two-step process has been previously known (see, for example, Ref. 6 Longenberger, L.; Mills, G. Formation of Metal Particles in Aqueous Solutions by Reactions of Metal Complexes with Polymers. J. Phys. Chem. 1995, 99, 475-478) and our data are consistent with such process. The data was re-analyzed and presented as per reviewer comments. Careful analysis of the data suggests that the Au3+ reduction to Au0 as observed by the rapid appearance of the peak at 240nm, which was detected as early as 1 min for reaction at 25oC as well as reaction at 45oC. No GNPs were formed at this time-point. The low temperature experiment at 4oC carried out under conditions when micelles do not form suggests the role of micelles in the GNPs formation. The formation of Au0 slowed down at 4oC, however, at sufficiently high Pluronic concentrations the reduction of Au3+ was still nearly complete yet the GNP formation was impaired. Likewise, we observed impaired GNPs when PEG was used instead of Pluronic. This reinforces the role of the micellization.

Size values given to the tenth of nm (61.6 ± 0.4 nm) do not make sense in view of the TEM images (S3 and S4 for instance). Such images would be more readable if tiled at identical scale.

The size values are Deff as measured by DLS and are indicative of the z-average particle diameter.  We change their presentation them to the nm. The TEM measurements were mostly performed using identical scale of 100nm, except when using smaller scale would result in mis-representing the particles shapes and sizes in population. In this case scale of 500 nm was used (Fig.S7).

Refs 19 and 31 are incomplete.

The references 19 and 31 were corrected. Thank you.

Reviewer 2 Report

This manuscript describes the gold nanoparticle synthesis in aqueous Pluronic solutions. In particular, authors examine the effect of temperature, pH, NaCl, [AuCl4]- concentration and purification on the [AuCl4]- reduction and gold nanoparticle formation to establish the controlled synthesis of gold nanoparticles in the Pluronic system. Furthermore, authors characterize the polymers before and after reaction using NMR, FTIR and TGA to evaluate the reaction mechanism of Pluronics with [AuCl4]-. Authors propose the mechanism on the [AuCl4]- reduction and gold nanoparticle formation by Pluronics and role of Pluronics for the [AuCl4]- reduction and gold nanoparticle formation in aqueous media. The findings obtained in this work should lead to the further development of the methodology for the gold nanoparticle synthesis using Pluronics. Therefore, the manuscript is worthy of publication in the polymers. To further clarify the paper, the authors are encouraged to address the following points:

1)      Au+3 should be modified to Au3+.

2)      Authors reveal the formation of free radicals and hydroperoxides after reaction. However, authors do not discuss the contribution of free radicals and hydroperoxides to the [AuCl4]- reduction and gold nanoparticle formation. Authors should describe the reaction schemes proposed and schematic illustration of mechanism on the [AuCl4]- reduction and gold nanoparticle formation by Pluronics.

3)      If the free radicals and hydroperoxides generated from Pluronics contribute to the [AuCl4]- reduction, the free radicals and hydroperoxides must be detected before reaction. Authors should discuss the formation of the free radicals and hydroperoxides before and after reaction in order to elucidate the role of the free radicals and hydroperoxides generated from Pluronics for [AuCl4]- reduction.

4)      In the references 7-9, it is described that the activity of [AuCl4]- reduction by Pluronics changes depending on the PEO and PPO chain length of Pluronics. Authors should explain this phenomenon in terms of formation of free radicals and hydroperoxides from Pluronics.

5)      Authors examine the effect of pH on the [AuCl4]- reduction and gold nanoparticle formation. Authors must discuss the connection between the state of [AuCl4]- depending on the pH, [AuCl4]- reduction and gold nanoparticle formation because [AuCl4]- changes to Au(OH)3 at base condition.

6)      Authors conclude that Pluronic is not adsorbed on the surface of gold nanoparticles formed. On the other hand, the organic compounds of ~13% are detected in the purified gold nanoparticles. Authors should show the components of the organic compounds in the gold nanoparticles formed. Also authors should describe the state and location of Pluronics in the gold nanoparticles or in the solutions.

7)      Authors detect the degradation compounds of Pluronic after reaction using NMR and FTIR. Authors should show the quantitative information (yield %) of the degradation compounds of Pluronic.

Author Response

Reviewer 2

This manuscript describes the gold nanoparticle synthesis in aqueous Pluronic solutions. In particular, authors examine the effect of temperature, pH, NaCl, [AuCl4]- concentration and purification on the [AuCl4]- reduction and gold nanoparticle formation to establish the controlled synthesis of gold nanoparticles in the Pluronic system. Furthermore, authors characterize the polymers before and after reaction using NMR, FTIR and TGA to evaluate the reaction mechanism of Pluronics with [AuCl4]-. Authors propose the mechanism on the [AuCl4]- reduction and gold nanoparticle formation by Pluronics and role of Pluronics for the [AuCl4]- reduction and gold nanoparticle formation in aqueous media. The findings obtained in this work should lead to the further development of the methodology for the gold nanoparticle synthesis using Pluronics. Therefore, the manuscript is worthy of publication in the polymers. To further clarify the paper, the authors are encouraged to address the following points:

Au+3 should be modified to Au3+.

This was corrected.

Authors reveal the formation of free radicals and hydroperoxides after reaction. However, authors do not discuss the contribution of free radicals and hydroperoxides to the [AuCl4]- reduction and gold nanoparticle formation. Authors should describe the reaction schemes proposed and schematic illustration of mechanism on the [AuCl4]- reduction and gold nanoparticle formation by Pluronics.

We appreciate the comments. The proposed mechanism is now added as new section in results, (“3.13 Mechanism of HAuCl4•3H2O reduction by Pluronic”, line 624) and further explained in discussion line 713.

If the free radicals and hydroperoxides generated from Pluronics contribute to the [AuCl4]- reduction, the free radicals and hydroperoxides must be detected before reaction. Authors should discuss the formation of the free radicals and hydroperoxides before and after reaction in order to elucidate the role of the free radicals and hydroperoxides generated from Pluronics for [AuCl4]- reduction.

We appreciate the comment. The levels of radicals and hydroxy-peroxides before the reaction were indeed analyzed and presented in Figure 10. We made changes in the discussion that clarify this point. Also, this point is clarified in the reaction scheme that we added in response to the previous critique.

4)      In the references 7-9, it is described that the activity of [AuCl4]- reduction by Pluronics changes depending on the PEO and PPO chain length of Pluronics. Authors should explain this phenomenon in terms of formation of free radicals and hydroperoxides from Pluronics.

We appreciate the comment. See the reaction scheme we propose for polymer degradation, in section 3.13 Mechanism of HAuCl4•3H2O reduction by Pluronic, line 624 and the reference.

Actually, more PPO more stable peroxides in PPO – hydroperoxides are more stable and likely to form in tertiary carbon atom.

Authors examine the effect of pH on the [AuCl4]- reduction and gold nanoparticle formation. Authors must discuss the connection between the state of [AuCl4]- depending on the pH, [AuCl4]- reduction and gold nanoparticle formation because [AuCl4]- changes to Au(OH)3 at base condition.

Thank you for this comment! At acidic conditions formation of ion radical •O2- and HO2- is inhibited, which should decrease the rate of reduction of Au3+. This is consistent with our observation that at low pH the reduction reaction was considerably decelerated. At alkali condition AuCl4 is transferred to Au(OH)3. We now point it out in the reaction scheme and discussion sections.

6)      Authors conclude that Pluronic is not adsorbed on the surface of gold nanoparticles formed. On the other hand, the organic compounds of ~13% are detected in the purified gold nanoparticles. Authors should show the components of the organic compounds in the gold nanoparticles formed. Also authors should describe the state and location of Pluronics in the gold nanoparticles or in the solutions.

We appreciate the suggestion. While it would be beneficial to determine the exact structure of the organic component on GNP surface, it is not possible to desorb this fraction, without affecting or degrade their structure. In our results we suggest that these are shorter PEO segments terminated with OH groups as confirmed by NMR and IR studies in Figure 8. This was clarified in the text, line 490. In the discussion we point out that that we believe that they are localized at the surface of GNPs.

Authors detect the degradation compounds of Pluronic after reaction using NMR and FTIR. Authors should show the quantitative information (yield %) of the degradation compounds of Pluronic.

We appreciate the comment. The Figure S7 was changed to better present the data and the yield of degradation was calculated by comparing the RI area for F127 associated peaks and the degradation products peak (55%, under optimal reaction conditions). The data was added to the results, line 531.

Round 2

Reviewer 1 Report

The manuscript has been improved in term of length and clarity and the authors have edited the unsupported claims. The manuscript still involves some cumbersome parts e.g. the purification of particles should not be a result but should be shortened and go into Materials and Methods (Figure 3A-B useless). Other changes could include a change in the scales font in Fig 3D (not readable) and of the photo scales: Indeed the scale bars all represent 100 nm, but they do not all have the same length ! In order to compare NP sizes, we need to use a ruler.

Author Response

We appreciate reviewers comments. 

The manuscript has been improved in term of length and clarity and the authors have edited the unsupported claims. The manuscript still involves some cumbersome parts e.g. the purification of particles should not be a result but should be shortened and go into Materials and Methods (Figure 3A-B useless).

This section was significantly shortened and combined with section 3.2 (lines 227-240). Figure 3A and B were removed. and Figures 3C and 3D (now A and B) quality was improved. 

Other changes could include a change in the scales font in Fig 3D (not readable) and of the photo scales: Indeed the scale bars all represent 100 nm, but they do not all have the same length ! In order to compare NP sizes, we need to use a ruler.

Figures 3C and 3D (now A and B) quality was improved and scale bar added to Fig 3D. 

Reviewer 2 Report

The manuscript was revised following the comments from reviewers.  The revised manuscript becomes clearer and emphasizes the novel findings. The manuscript is recommended for publication in the polymers.

Author Response

We appreciate reviewers effort and positive comment.